# Adolescents Who Play and Spend Money in Simulated Gambling Games Are at Heightened Risk of Gambling Problems

**DOI:** 10.3390/ijerph191710652

**Published:** 2022-08-26

**Authors:** Nerilee Hing, Cassandra K. Dittman, Alex M. T. Russell, Daniel L. King, Matthew Rockloff, Matthew Browne, Philip Newall, Nancy Greer

**Affiliations:** 1Experimental Gambling Research Laboratory, School of Health, Medical and Applied Sciences, Central Queensland University, Bundaberg 4670, Australia; 2College of Education, Psychology, and Social Work, Flinders University, Adelaide 5042, Australia; 3School of Psychological Science, University of Bristol, Bristol BS8 1TU, UK

**Keywords:** social casino games, demo games, gaming, video games, gambling disorder, microtransactions, youth, young people

## Abstract

Simulated gambling, such as playing a virtual slot machine for points rather than money, is increasingly part of the online gaming experience for youth. This study aimed to examine (1) if youth participation in simulated gambling games is associated with participation in monetary gambling; (2) if youth participation in simulated gambling games is associated with increased risk of problematic gambling when controlling for breadth of monetary gambling (i.e., number of gambling forms); and (3) if monetary expenditure and time spent playing simulated gambling games increase the risk of problematic gambling. Two samples of Australians aged 12–17 years were recruited—826 respondents through an online panel aggregator (mean age 14.1 years) and 843 respondents through advertising (mean age 14.6 years). Aim 1 was addressed using chi-square and correlation analyses. Linear multiple regression analyses were conducted to address Aims 2 and 3. The findings in both samples supported the study’s hypotheses—that (1) youth who play simulated gambling games are more likely to participate in monetary gambling, and that (2) participation and (3) time and money expenditure on simulated gambling are positively and independently associated with risk of problematic gambling when controlling for the number of monetary gambling forms, impulsivity, age and gender. To better protect young people, simulated gambling should, at minimum, emulate the consumer protection measures required for online gambling.

## 1. Introduction

Rapid changes in technology have heralded a convergence of gaming and gambling. Video games played by young people are now replete with gambling elements. Simulated gambling games are increasingly part of the online gaming experience [1]. These simulated games include, for example, slot machines that can be played for points instead of for real money; legally and with limited accompanying consumer advice. These developments have led to calls for further research on emerging gambling products [2] given the gaps in knowledge around the role of technology in shaping gambling behaviours [3]. The research problem of central interest in the current paper is whether young people who participate and spend money in simulated gambling games face an elevated risk of gambling problems.

Estimated past-year prevalence of problem gambling in adolescents ranges from 0.2% to 5.6% across various jurisdictions, although comparability of these figures is compromised by methodological variations [4]. Arguably, the most population-representative recent studies have been conducted in the United Kingdom, reporting a prevalence of 1.7% in both 2018 and 2019 [5,6]. Similar prevalence rates have recently been reported in Australia [7,8], where 1.4–1.5% of adolescents meet criteria for problem gambling and a further 2–7% are in the at-risk category when measured on the DSM-IV-MR-J [9]. Gambling problems are at least as prevalent, if not more so, amongst adolescents than adults [10]. While young people are unlikely to experience the same degree of financial harm as adults, their gambling nevertheless can have negative consequences for their education, mental health, family and social relationships, and pose an increased risk for the development of adult gambling problems [2,11,12]. Given these consequences, it is important to understand whether participation in simulated gambling games increases the risk of problem gambling amongst young people.

### 1.1. Types and Prevalence of Simulated Gambling

Simulated gambling in online games takes many forms. Many video games have ‘mini’ gambling components. In these games, gambling is not the game’s central feature, but rather gambling simulations are embedded in parts of gameplay. Examples of these embedded games include: big-wheel, slot machines and casino card games [13]. Players may need to engage in these discrete activities to progress in games, gain lives, earn in-game currency or obtain free items [1,8,14]. In contrast, social casino games played on gambling-themed apps or social networking sites replicate gambling games as the central feature of play. While social casino games can be played for free, most allow players to purchase virtual credits with real money but reward players only with virtual currency or points [13]. Demo or practice games are a further variation of nonmonetised simulated gambling that are provided on real gambling websites or apps to enable players to try these games for free [15,16]. Simulated gambling more typically involves financial options, such as the option to purchase virtual currency to continue play when the freely available credit has been exhausted.

A substantial proportion of adolescents play simulated gambling games, with higher rates of participation amongst males [6,8]. Around two-fifths of adolescents report past-year participation in simulated gambling games [8,17,18]. The most commonly played are games with mini gambling components, by 30–40% of adolescents [8,17,18]. About 12–25% of adolescents reportedly play social casino games, and 8–20% play demo games [6,8,19].

### 1.2. Research into Simulated Gambling, Monetary Gambling and Gambling Problems amongst Adolescents

Research has started to explore whether simulated gambling increases gambling and gambling problems amongst young people. It has been proposed that simulated gambling may provide a ‘gateway’ to gambling and later gambling problems [1,20,21,22]. Cross-sectional studies have found that simulated gambling is more prevalent amongst young people who gamble but cannot yet identify causal directions [8,14,23,24]. Longitudinal studies have found some migration from simulated to monetary gambling. Dussault et al. [25] recruited 1220 adolescents who had never gambled, finding that 28.8% reported gambling at one-year follow-up. However, simulated gambling at baseline predicted later uptake of monetary gambling only in relation to card-based poker. Amongst 1178 school students [17], 11.9% of those who had never gambled nevertheless reported gambling one year later, and this migration was associated with prior use of simulated gambling on social networks. Further analysis found that simulated gambling impacted problem gambling mostly via the indirect effects of gambling frequency and irrational cognitions, with problematic internet gaming being a decreasing mechanism for problem gambling [26]. Cross-sectional correlational research has also found that young people with gambling problems have higher participation in simulated gambling games [14,19,24,27]. However, previous studies examining relationships between simulated gambling and problem gambling have relied on small subsamples of at-risk/problem gamblers, have not always used a validated measure of problem gambling, have lowered the cutoff score of validated instruments to have sufficient numbers for analysis, or have included only one type of simulated gambling. The current study was designed to overcome these limitations.

### 1.3. Features of Simulated Gambling Thought to Increase the Risk of Gambling Problems

Longitudinal research has not yet been conducted to provide further evidence for a potential causal link between simulated gambling and gambling problems amongst youth. Nonetheless, researchers have proposed several psychosocial mechanisms by which simulated gambling may increase engagement in gambling and the risk of problematic gambling [1]. Simulated gambling may normalise gambling and increase youth exposure to a gambling subculture and peer pressure to gamble, which in turn could contribute to it being perceived as a safe and socially acceptable activity [28]. Simulated gambling also allows players to practise and experiment with gambling, which can build familiarity, confidence and an inflated sense of skill that can elevate risk-taking in monetary gambling [16,22,29,30]. However, the inflated odds and early big wins in simulated gambling games may lead to the explicit or implicit perception that winning at gambling is easy and success results from use of strategy and practice, thereby nurturing overconfidence in the probability of winning in monetary gambling [22,31,32,33]. Simulated gambling may also expose adolescents to direct links to gambling opportunities and advertising, which promote gambling as a fun, glamorous and exciting activity [22,29]. Another risk is that the gambling features in online games may make video-gaming more problematic or addictive for some players [28,29], which may lead to further engagement in simulated gambling and monetary gambling. The randomised and occasionally manipulated rewards of simulated gambling mean that the amount of gameplay needed to win is unknown, which may nurture behavioural conditioning and persistence [1,34].

Researchers have also proposed that simulated gambling encourages real-money expenditure that may become excessive. Online games have increasingly become monetised, with players encouraged to pay for credits, boosts, faster speeds, continued play and in-game items [1]. Parent Zone [35] found that 76% of adolescents reported games often try to get them to spend money, and almost half (49%) reported that online games are only fun when they do spend money. This forms part of a broader trend of gaming monetisation in which game access is provided for free, but increasingly large spends are required for more intensive play. Thus, a ‘long tail’ of consumers, who become highly engaged with the game, contribute disproportionately to the revenue of the provider, who is in turn incentivised to maximise engagement. Parent Zone [35] also observed that some games are practically unplayable without spending money, and that psychological manipulation to encourage spending uses tactics based on loss aversion, reward removal and variable-ratio reinforcement (see [36]). Armstrong et al. [29] also explain that a focus on game outcomes rather than losses of virtual currency, along with the lower psychological value of virtual money, are detrimental determinants of behaviour if transferred to monetary gambling. The normalisation of large bet sizes, pay-to-win options and frequent bonus credits in simulated gambling may also increase the likelihood of excessive expenditure in monetary gambling [22].

King and Delfabbro [28] further note that players may overspend in monetised games due to ‘entrapment’, believing they have invested too much to quit. While purchasing loot boxes is associated with increased gambling problems amongst youth [34,37,38,39], only one study has examined links between spending money on other simulated gambling games and problematic gambling in adolescents [27]. In that study, 64.5% of adolescents who had paid when playing social casino games reported gambling for real money as a result of playing these games, compared to only 1.3% of nonpayers. Adolescents engaging in microtransactions also reported more frequent participation and spending on monetary gambling and more symptoms of problem gambling.

### 1.4. The Current Study

Overall, the potentially risky features of simulated gambling, the popularity of gaming amongst young people, and their vulnerability to gambling problems, indicate the importance of understanding whether simulated gambling elevates their risk of problematic gambling. Therefore, the aims of this study were to examine (1) if youth participation in simulated gambling games is associated with participation in monetary gambling; (2) if youth participation in simulated gambling games is associated with increased risk of problematic gambling when controlling for breadth of monetary gambling (i.e., number of gambling forms), impulsivity, age and gender; and (3) if monetary expenditure and time spent playing simulated gambling games increase the risk of problematic gambling. Although previous research findings are limited, as discussed above, they support our hypothesised relationships—that (1) youth who play simulated gambling games are more likely to participate in monetary gambling, and that (2) participation and (3) time and money expenditure on simulated gambling are positively and independently associated with the risk of problematic gambling. The study will contribute to knowledge by testing these hypotheses, which in turn can inform public health measures to protect young people from any risks to their wellbeing associated with playing simulated gambling games.

## 2. Materials and Methods

Our institutional ethics board approved the study, which entailed an online survey of two samples of adolescents in Australia, who were under the legal gambling age of 18 years.

### 2.1. Sampling, Recruitment and Participants

Survey inclusion criteria were being aged 12–17 years, residing in New South Wales Australia (NSW), permission from a parent/guardian to participate, and the adolescent’s informed consent. We recruited nonprobability samples to ensure enough respondents with symptoms of problematic gambling for the planned analyses. Because the analyses test relationships between variables, representative samples are inefficient and not essential [40]. Nonetheless, we purposefully recruited the samples using very different means to enhance confidence in the generalisability of any results that were found to be consistent across both samples. Table 1 displays the demographic characteristics for both samples.

*Sample 1*. We recruited 826 respondents through Qualtrics, which sources respondents from multiple online panels. Safeguards for data quality included de-duplicating responses through checking IP address and similarity in responses. Respondents were removed who failed attention check questions, straight-lined through responses, sped through the survey, gave inconsistent responses, or nonsense responses to open-ended questions. Two researchers confirmed the exclusion decisions. Of the 1098 eligible people who started the survey, we excluded 42 responses that failed these data quality checks. Of the remaining eligible 1056 participants, 826 completed the survey for a completion rate of 78.2%. These respondents were compensated by Qualtrics based on their internal points-accumulation system. The survey ran from 16 April to 5 May 2020. Just over half of participants were male (55.1%) with a mean age of 14.81 years (*SD* = 1.64) (see Table 1). As expected in panel samples [40], rates of past-year problem (15.5%) and at-risk (9.2%) gambling were higher than in population studies, with lower rates of nonproblem gambling (24.6%) and nongambling (50.7%).

*Sample 2*. We recruited another 843 participants through email and online advertising. We emailed all our laboratory’s previous gambling research participants who resided in NSW and had consented to receive invitations for future research. The email invitation requested they ask any adolescents in their household if they would like to complete the survey. The survey was also advertised for two weeks on Facebook, Instagram and Twitter, and via the funding agency’s online communications channels. To identify any duplicate responses, we examined the email addresses and unique codes for follow-up that respondents provided. We also examined IP addresses, noting that respondents from the same household could complete the survey, and that some IP duplication was therefore possible. We found no evidence of wide-scale or suspicious duplications that warranted removal of responses. In total, 1404 eligible people started the survey and 841 completed it, for a completion rate of 60.0%. Respondents could enter a survey prize draw for a gift voucher valued at AU$100. The survey ran from 23 April to 11 May 2020. Most participants were male (69.3%) with a mean age of 14.6 years (*SD* = 1.7) (see Table 1). As expected, rates of past-year problem (49.9%) and at-risk (8.2%) gambling were higher than those found in representative studies, with lower rates of nonproblem gambling (10.9%) and nongambling (31.0%). Respondents from our previous gambling studies, and those on gambling-related mailouts or targeted by social media advertising, are likely to be more engaged gamblers. Given the strong link between parental and youth gambling problems [41], this sample was expected to have a higher prevalence of gambling problems when compared to sample 1.

### 2.2. Measures

The survey instrument and details of its cognitive testing are available in (blinded for anonymity). Measures used in the current paper are briefly described below.

*Problematic gambling* in relation to monetary gambling was assessed using the DSM-IV-MR-J [9]. It consists of nine questions (e.g., ‘During the last 12 months, how often have you found yourself thinking about gambling or planning to gamble?’). For descriptive purposes, respondents who endorsed 4 or more items were classified as experiencing gambling problems, 2 to 3 items were classified as at-risk, and 0 to 1 item as not experiencing problems. Inferential analyses detailed below used the summed DSM-IV-MR-J scores as a continuous scale, which could range from 0 to 9. Cronbach’s alpha was 0.80.

*Participation in monetary gambling* was assessed for 11 activities (see Table 2). Response options were: ‘In the last 7 days’, ‘In the last 4 weeks’, ‘In the last 12 months’, ‘More than 12 months ago’ or ‘Never’. Respondents could only select one of these options to reflect when they had most recently participated in each activity. Past-month participation in each form of monetary gambling was analysed by combining the first two response options (coded 1) and other options (coded 0).

*Breadth of monetary gambling* was calculated for some analyses, as indicated below, by summing the count of monetary gambling forms the respondent had participated in during the past month.

*Simulated gambling games.* This survey section had the following preamble: ‘Games have gambling components, which look and play like normal gambling games—for example roulette, poker, slot machines and bingo—such as those shown below [several images were included]. They may be free to play, or you may pay to play, but you cannot win real money’. Respondents were then asked: ‘When, if ever, did you last play any of these games with gambling components?’ with four types listed: ‘Video games with gambling components (such as Diamond Casino & Resort in the video game Grand Theft Auto V)’; ‘Gambling-themed apps from an app store (such as bingo, poker, pokies/slots, or roulette that you play on your phone, tablet or computer)’; ‘Games with gambling components on social networking websites (such as Zynga games on Facebook)’; and ‘Free demo or practice games on real gambling websites or apps, for example, Mobile Casinos’. Response options were: ‘In the last 7 days’, ‘In the last 4 weeks’, ‘In the last 12 months’, ‘More than 12 months ago’ or ‘Never’. Past-month participation in each type of simulated gambling game was analysed by combining the first two response options (coded 1) and remaining options (coded 0).

*Time spent playing simulated gambling games*. Respondents who reported having played games with gambling components were asked: ‘In general, about how many hours per week OR per month do you usually spend playing games with gambling components?’ Responses were recoded before analysis to hours per month.

*Monetary expenditure on simulated gambling games*. Respondents who reported having played games with gambling components were asked: ‘In a typical month, about how much do you spend on microtransactions, such as to get virtual credits, in games with gambling components (not including loot boxes)?’. Responses were provided in Australian dollars (AUD).

*Impulsivity* was assessed using the Barratt Impulsiveness Scale—Brief [42]. The BIS-Brief contains eight items (e.g., ‘I plan tasks carefully’). After reverse-coding appropriate items, items were summed for a total score. Cronbach’s alpha was 0.71.

*Demographic characteristics* included in the analysis were age (measured in years) and gender (male, female, other).

### 2.3. Statistical Analysis

All analyses were conducted using IBM SPSS Statistics for Windows, version 28. Separate analyses were conducted for the Qualtrics sample and Emails/Advertisements sample, respectively, due to their differing sampling characteristics. To assess the first research question, a series of chi square analyses were conducted to examine whether past-month participation in each of the simulated gambling forms was associated with past-month participation in each of the 11 forms of monetary gambling. Moreover, correlation analyses were used to assess the relationship between each of the simulated gambling forms and the total number of monetary forms of gambling participated in during the past month (i.e., breadth of monetary gambling).

To assess the second research question, linear multiple regression analyses were conducted to assess if past-month adolescent participation in each of the simulated gambling forms was associated with problematic gambling symptoms (i.e., DSM-IV-MR-J scores) while controlling for adolescent age, gender, impulsivity (BIS scores), and breadth of monetary gambling in the past month (i.e., count of gambling forms). Separate regression models were evaluated to first assess the contribution of the key covariates of age, gender and impulsivity to problematic gambling (model 1); the added contribution of count of monetary gambling forms (model 2); and the contribution of each simulated gambling form entered separately (models 3 to 6). Models 3 to 6 were intended to evaluate the added risk to gambling problems implied by participation in these simulated forms of gambling over-and-above risk factors evaluated in the prior models.

The third research question was addressed using linear multiple regression analyses on the restricted set of participants within each sample who identified as having participated in at least one form of simulated gambling in the past 12 months (Qualtrics sample *N* = 339; Emails/Advertisements sample *N* = 636). The regression analyses examined whether time spent on simulated gambling each month and monetary expenditure (AUD) on these games each month were each independently associated with problematic gambling, after controlling for adolescent age, gender and impulsivity (BIS scores). Following a similar entry sequence as for research question 2 above, four sets of analyses were conducted with each sample. Model 1 evaluated the covariates only; model 2 assessed the added contribution of expenditure in simulated gambling; model 3 evaluated the contribution of time spent in simulated gambling; and model 4 assessed the independent contributions of both monetary expenditure and time spent in simulated gambling.

## 3. Results

### 3.1. Past-Month Participation in Simulated Gambling and Participation in Monetary Gambling

Table 2 describes the results of chi-square analyses investigating the relationship between adolescents’ participation in simulated gambling activities in the past month with their participation in monetary gambling activities in the past month. In the Qualtrics sample of participants, there were consistent associations between each simulated gambling activity and each monetary gambling activity. Within the Email/Advertisements sample, the use of social casino games on social networking sites was associated with every form of monetary gambling, as was participation in free demo games on gambling websites. In comparison, adolescents’ use of games with ‘mini’ gambling components was related to every monetary gambling activity except for betting on Keno, poker and casino table games. Participation in social casino games via apps was related to monetary gambling on poker machines, scratchies or lottery, bingo or housie, poker, esports, fantasy sports and informal gambling, but not to betting on horse or greyhound races, Keno, casino table games or sporting events. Finally, across both samples, participation in simulated gambling in the past month was associated with participation in a greater number (count) of monetary gambling activities overall.

### 3.2. Past-Month Simulated Gambling as a Risk Factor for Problematic Gambling Symptoms

#### 3.2.1. Assumptions Checking

Assessment of the assumptions of linear regression revealed no violation of the assumptions of homoscedasticity, multicollinearity and independence of observations. However, distributional checks for violations of normality revealed that the distribution of the DSM-IV-MR-J in the Qualtrics sample was positively skewed, as were the distributions of participation in monetary gambling in both samples. In addition, there were several univariate outliers on the DSM-IV-MR-J in the Qualtrics sample and on participation in monetary gambling in both samples. When appropriate transformations to reduce the influence of the outliers and skew were applied (i.e., log and trim), there were no substantive changes to the outcome of the regression analyses. Thus, the original, nontransformed scores were retained in all analyses.

#### 3.2.2. Results of Analyses

Table 3 reports the results of multiple regression analyses predicting problematic gambling symptoms from each simulated gambling activity, after controlling for adolescent age, gender, impulsivity, and total number of monetary gambling activities in the past month (Qualtrics sample *N* = 826; Emails/Advertisements sample *N* = 843). The results of Model 1 show the contribution of the covariates of age, gender and impulsivity, Model 2 shows the additional contribution of the number of monetary gambling activities, while Models 3 to 6 show the contribution of each simulated gambling activity entered individually into the model after controlling for the other variables.

Across both samples, adolescent impulsivity and breadth of monetary gambling (i.e., count of forms) reliably and independently predicted adolescent problematic gambling scores. Adolescent age (but not gender) was a unique predictor within the Qualtrics sample, but not the Email/Advertisements sample. Breadth of monetary gambling was the strongest predictor of problematic gambling scores, accounting for 39% of unique variance in problematic gambling in the Qualtrics sample, and 55% of variance in the Email/Advertisements sample, after controlling for age, gender and impulsivity.

Despite the very strong contribution made by breadth of monetary gambling, participation in various simulated gambling forms still made a small but unique additional contribution to the prediction of problematic gambling symptoms. Specifically, in the Qualtrics sample, each simulated gambling form was independently associated with problematic gambling scores, while in the Email/Advertisements sample, all games except for those with ‘mini’ gambling components independently predicted problematic gambling scores. Thus, within a highly controlled model, adolescent participation in simulated gambling over the past month was associated with increased problematic gambling symptoms, even after controlling for adolescent age, gender, impulsivity and breadth of monetary gambling.

### 3.3. Monetary Expenditure and Time Spent Playing on Simulated Gambling Games as Risk Factors for Gambling Problems

#### Assumptions Checking

Assessment of the assumptions of linear regression revealed no violation of the assumptions of homoscedasticity, multicollinearity, and independence of observations. However, both hours spent playing simulated gambling games and money spent on simulated gambling per month were significantly positively skewed in both samples, and there were a number of univariate outliers. A logarithmic transformation was conducted on the scores on these variables to reduce the influence of the outliers and improve the normality of their distributions. When regression analyses were run with and without these transformed variables, the model fit was substantially improved with the transformed variables. Thus, the transformed variables were retained in all analyses across both samples.

### 3.4. Results of Analyses

Table 4 displays the results of regression analyses conducted using the subsample of participants who reported engaging in at least one form of simulated gambling in the past 12 months (Qualtrics sample *N* = 339; Emails/Advertisements sample *N* = 636). These analyses examined the contribution of monetary expenditure and time spent (hours) on simulated gambling per month to the prediction of problematic gambling symptoms.

None of the control variables (i.e., adolescent age, gender and impulsivity) reliably predicted problematic gambling scores across samples (model 1). Monetary expenditure was the only significant predictor in both samples when it was entered alongside the control variables (model 2). When time spent on simulated gambling was substituted for monetary expenditure (model 3), youth impulsivity was a significant predictor of problematic gambling in both samples, while older age was related to problematic gambling in the Qualtrics sample, and male gender was associated with problematic gambling in the Email/Advertisements sample. However, these associations were not significant in models containing monetary expenditure as a predictor.

## 4. Discussion

The findings supported the study’s hypotheses—that (1) youth who play simulated gambling games are more likely to participate in monetary gambling, and that (2) participation and (3) time and money expenditure in simulated gambling are positively associated with risk of problematic gambling.

### 4.1. Past-Month Simulated Gambling Related to Past-Month Gambling Participation

First, adolescents who engaged in simulated gambling games in the past month were more likely to also participate in past-month monetary gambling. In both samples, past-month participation in social casino games on apps or social networking sites, demo games on gambling operator sites, and games with ‘mini’ gambling components were each associated with past-month participation in all forms of monetary gambling in the Qualtrics sample, and the majority of gambling forms in the Email/Ads sample. Adolescents who participated in each type of simulated gambling also tended to have greater breadth of monetary gambling, that is, to participate in a greater diversity of gambling forms. These results were consistent with previous findings that monetary gambling is more prevalent amongst young people who play these types of simulated gambling games [8,14,17,23,24]. The findings present new evidence that reflects a convergence between gaming and gambling behaviours during adolescence, which is also found amongst young people who participate in other gamblified activities linked to video games, including loot box purchasing, esports betting and skin gambling [34,37,43,44,45,46,47].

### 4.2. Past-Month Simulated Gambling Uniquely Predicts Gambling Problems

Second, adolescents with past-month participation in simulated gambling reported more problematic gambling symptoms, when controlling for the number of past-month monetary gambling forms, impulsivity, age and gender. Across both samples, the number of past-month monetary gambling forms had the strongest association with increased risk of problem gambling, while older age and higher impulsivity also explained some unique variance. Further, participation in each of the four simulated gambling games also made small but independent contributions to an increase in problematic gambling symptoms, except for games with ‘mini’ gambling components in the Email/Ads sample. Thus, both samples indicated that playing social casino games or demo games made a unique contribution to increased risk of problematic gambling. These results pertain to a broader range of simulated gambling games than most prior studies have examined, and align with previous findings that adolescents with a gambling problem are more likely to participate in simulated gambling games that closely replicate monetary gambling games [14,19,24,26,27], as well as other gambling activities linked to video games, including purchasing loot boxes, skin gambling and esports betting [37,39,43,44,46,47,48].

Naturally, this cross-sectional study does not indicate causality between simulated gambling, monetary gambling and problematic gambling. It is speculated that simulated gambling may encourage gambling and harmful gambling through the psychosocial processes discussed earlier, including normalisation, familiarisation, confidence-building, heightened expectations of winning, and behavioural conditioning [1,22,28,29]. Supporting this temporal sequence, two longitudinal studies with adolescents both found evidence that prior participation in simulated gambling was associated with later gambling uptake [17,25]. Further, an experimental study with adults provided causal evidence linking simulated gambling to subsequent monetary gambling [48]. Amongst participants randomly assigned to play a bespoke simulated slots game in a 24-week trial, the number of simulated gambling sessions played in one week predicted real-money slots play in the subsequent week. There is a need for experimental and longitudinal studies to investigate the potential causal relationship between simulated gambling and gambling problems in youth.

An alternative explanation is that simulated gambling may provide a substitute activity when gamblers are unable to gamble or want to limit their monetary gambling, especially if it is causing harm. However, studies investigating this possibility have either found no evidence of this effect [48] or that it is reported by very few participants [32,49], suggesting that simulated gambling is more likely to act as a ‘catalyst’ rather than a ‘containment’ strategy for real-money gambling (see [22]). A third variable explanation is also possible. Given that simulated gambling games emulate their monetary equivalents, it is not surprising that they might both appeal to particular types of young people, such as those who are older or more impulsive, as found in the current study. Several studies have found that older age and higher impulsivity are risk factors for greater breadth of gambling involvement and gambling problems amongst youth [4,50,51,52,53]. Other variables not measured in the current study may explain the appeal of both simulated and monetary gambling to higher-risk adolescent gamblers.

### 4.3. Time and Money Spent per Month on Simulated Gambling Independently Related to Gambling Problems

The third finding is that monetary expenditure and time spent per month on simulated gambling were positively associated with the risk of problematic gambling. In both samples, expenditure through microtransactions and time spent on simulated gambling were both significantly associated with problematic gambling symptoms after controlling for age, gender and impulsivity. In fact, age, gender and impulsivity were not significantly associated with problematic gambling scores once money and time expenditure were included in the model. Money spent was a stronger predictor than time spent in both samples. The results add further evidence to support previous findings that young people who spend real money in social casino games are significantly more likely to report problem gambling symptoms [27]. Elevated problem gambling rates have also been found amongst adolescents who spend money on loot boxes in digital games [34,37,38,39]. The association between expenditure in games and problem gambling is of public health concern given the increasing monetisation of digital games, which provide opportunities to spend real money and use exploitative practices to encourage repeated expenditure [54,55,56]. Not only does simulated gambling disproportionately attract adolescents with greater vulnerability to problem gambling, but real-money expenditure in these games may also worsen their financial situation that is already harmed by gambling.

## 5. Limitations

Overall, this study has added to the limited available research evidence on links between simulated gambling and monetary gambling amongst adolescents and is the first such study conducted amongst Australian youth that is inclusive of all major forms of simulated gambling. Clearly, more research is needed to clarify any causal directions between engagement in simulated gambling and monetary gambling, as well as with problematic gambling. The current study was limited by its cross-sectional design, which constrained the analyses to statistical associations. It was also limited through its reliance on self-report data, which may have introduced recall, social desirability and other biases. The two samples were nonprobability samples, which may not be representative of the adolescent population. However, the sampling approaches yielded sufficiently large subsamples of young people with problem gambling symptoms for the planned analyses to proceed. Given the low prevalence of gambling problems amongst youth, very large representative samples would be needed to conduct these analyses. Despite our use of two different sampling methods, the results were very consistent across the two samples, and aligned with previous research findings.

### Practical Implications

The results of this study indicate that young people who play simulated gambling games are more likely to gamble and to experience symptoms of problem gambling, which are further elevated amongst those who spend real money in these games. Young people who play simulated gambling games, and their parents, should therefore be an appropriate target for educational initiatives in schools and through other public health measures. These initiatives should aim to raise awareness of gambling problems, and how simulated gambling can reinforce harmful gambling behaviours, and elevate financial harm and other harms. However, adolescents have reported little parental supervision over their simulated gambling activities, indicating that countermeasures, including regulation, are also needed to limit access to these activities and their potential harm to young people [57].

At present, simulated gambling in digital games emulates the attractive features, structural characteristics and game-play mechanics of monetary gambling, but ignores measures required in digital monetary gambling to improve consumer protection and reduce harm. To better protect young people, simulated gambling should, at minimum, emulate the consumer protection measures required for online gambling. These should include age restrictions on access, restrictions on advertising, transparency about game mechanics and the odds of winning, the ability to track time and money expenditure, set limits, take breaks and self-exclude, and provide access to self-help resources and help service information. While these types of regulated measures for online gambling are modest and limited by their basis in an informed choice model of responsible gambling, they nonetheless provide some options to help consumers self-regulate their gambling. Introducing these measures for simulated gambling would provide tools to help young people self-regulate their involvement in these games. However, more stringent measures, such as a ban on simulated gambling elements in digital games played by children and young people, would better prevent them from engaging in these gambling-like activities before they reach legal gambling age.

## 6. Conclusions

At present, harm minimisation measures are required for monetary gambling that is regulated for adult use, but not for simulated gambling that is widely available to minors. The main difference between the two activities is that real money can be won in monetary gambling but not in simulated gambling, although real money can be expended on both activities. There is a need to introduce harm minimisation measures in simulated gambling games, given the many features of gambling they emulate that aim to encourage continued play and repeated expenditure. For these reasons, a preventive approach is warranted for youth engagement in simulated gambling games. This approach is particularly important given the steady growth of gamblified games, their popularity amongst youth, and their association with underage gambling and gambling problems in young people.

## Figures and Tables

**Table 1 ijerph-19-10652-t001:** Demographic characteristics of each sample.

	Qualtrics *N* = 826		Email/Advertisements *N* = 843	
	*M (SD)*	Range	*M (SD)*	Range
Adolescent age (years)	14.81 (1.64)	12–17	14.61 (1.66)	12–17
Total monetary gambling activities in past month	1.06 (2.19)	0–11	2.12 (2.06)	0–11
Hours spent in simulated gambling in past month	9.38 (27.15)	0–500	13.20 (17.55)	0–160
Expenditure in simulated gambling in past month (AUD)	13.09 (51.75)	0–1000	150.44 (201.43)	0–3000
Hours spent in simulated gambling in past month (log)	0.43 (0.32)	0–2.70	0.86 (0.57)	0–2.21
Expenditure in simulated gambling in past month (log)	0.38 (0.66)	0–3	1.40 (1.11)	0–3.48
Impulsivity	17.74 (4.40)	8–32	18.97 (4.03)	8–11
	*n*	%	*n*	%
Adolescent gender				
Female	370	44.8	258	30.6
Male	455	55.1	584	69.3
Other	1	0.1	1	0.1
Language spoken at home				
English	776	93.9	834	98.9
Language other than English	50	6.1	9	1.1
Parents’ living situation				
Living together	637	77.1	534	63.3
Not living together	189	22.9	309	36.7
Indigenous status				
Not Aboriginal or Torres Strait Islander	742	89.8	440	52.2
Aboriginal	40	4.8	171	20.3
Torres Strait Islander	8	1.0	122	14.5
Both Aboriginal and Torres Strait Islander	4	0.5	94	11.2
Prefer not to say	32	3.9	16	1.9
Paid employment				
Not in paid employment	580	70.2	725	86.0
Part-time/casual employment	227	27.5	104	12.3
Full-time employment	19	2.3	14	1.7
Problem gambling status				
Nongambler	419	50.7	261	31.0
Nonproblem gambler	203	24.6	92	10.9
At-risk gambler	76	9.2	69	8.2
Problem gambler	128	15.5	421	49.9
Participation in past-month simulated gambling				
Games with ‘mini’ gambling components	146	17.7	276	32.7
Social casino games via apps	129	15.6	289	34.3
Social casino games on social networking	116	14.0	309	36.7
Free demo games on gambling websites	118	14.3	193	22.9

**Table 2 ijerph-19-10652-t002:** The relationship between past-month simulated gambling and participation in past-month monetary gambling in each sample.

	Games with ‘Mini’ Gambling Components		Social Casino Games via Apps		Social Casino Games on Social Networking Sites		Free Demo Games on Gambling Websites	
Categorical Variables ^a^ (Ref = No)	Qualtrics Odds Ratio (95% CI)	Email/ads Odds Ratio (95% CI)	Qualtrics Odds Ratio (95% CI)	Email/ads Odds Ratio (95% CI)	Qualtrics Odds Ratio (95% CI)	Email/ads Odds Ratio (95% CI)	Qualtrics Odds Ratio (95% CI)	Email/ads Odds Ratio (95% CI)
Poker machines	8.47 *** (5.04; 14.25)	1.46 * (1.09; 1.97)	7.79 *** (4.63; 13.10)	4.84 *** (3.57; 6.58)	10.67 *** (6.29; 18.11)	3.68 *** (2.73; 4.95)	8.98 *** (5.31; 15.18)	4.14 *** (2.95; 5.79)
Horse or greyhound races	7.26 *** (4.46; 11.80)	6.54 *** (3.14; 13.63)	6.51 *** (3.99; 10.64)	1.69 (0.88; 3.22)	7.27 *** (4.42; 11.95)	13.08 *** (5.06; 33.82)	7.06 *** (4.30; 11.61)	3.44 *** (1.80; 6.59)
Scratchies, lottery, lotto, pools	4.66 *** (3.14; 6.91)	1.47 * (1.07; 2.03)	4.63 *** (3.08; 6.97)	2.83 * (2.07; 3.89)	5.02 *** (3.29; 7.64)	3.78 *** (2.74; 5.20)	5.56 *** (3.65; 8.45)	2.31 *** (1.64; 3.26)
Keno	7.95 *** (4.57; 13.84)	2.33 (0.94; 5.80)	8.87 *** (5.08; 15.48)	1.41 (0.56; 3.53)	8.19 *** (4.69; 13.31)	5.02 *** (1.79; 14.08)	5.78 *** (3.31; 10.10)	4.85 *** (1.92; 12.24)
Bingo or housie	6.14 *** (3.87; 9.74)	1.89 *** (1.37; 2.59)	7.16 *** (4.48; 11.45)	3.27 *** (2.38; 4.50)	7.22 *** (4.48; 11.63)	3.54 *** (2.57; 4.89)	6.61 *** (4.11; 10.63)	2.56 *** (1.81; 3.60)
Poker	6.48 *** (3.56; 11.81)	2.09 (0.86; 5.09)	7.75 *** (4.23; 14.18)	2.96 * (1.20; 7.32)	6.78 *** (3.70; 12.43)	3.31 ** (1.31; 8.38)	8.01 *** (4.37; 14.68)	4.31 *** (1.76; 10.55)
Casino table games	8.16 *** (4.21; 15.82)	2.39 (0.86; 6.65)	10.93 *** (5.58; 21.41)	2.23 (0.80; 6.20)	8.04 *** (4.17; 15.50)	11.68 *** (2.62; 52.12)	11.04 *** (5.66; 21.54)	9.76 *** (3.07; 31.02)
Sporting events	6.85 *** (4.07; 11.54)	6.01 *** (2.14; 16.85)	7.74 *** (4.57; 13.09)	2.17 (0.87; 5.40)	6.83 *** (4.02; 11.61)	6.76 *** (2.22; 20.56)	8.91 *** (5.24; 15.15)	3.89 ** (1.56; 9.72)
Esports	5.21 *** (3.22; 8.44)	1.62 ** (1.21; 2.19)	7.65 *** (4.69; 12.48)	4.01 *** (2.96; 5.43)	7.58 *** (4.61; 12.40)	3.80 *** (2.81; 5.13)	8.34 *** (5.09; 13.68)	2.83 *** (2.08; 4.03)
Fantasy sports games	7.90 *** (4.68; 13.34)	1.46 * (1.09; 1.96)	8.94 *** (5.27; 15.17)	4.03 *** (2.98; 5.45)	9.16 *** (5.38; 15.58)	3.37 *** (2.50; 4.52)	8.28 *** (4.87; 14.07)	2.52 *** (1.12; 3.50)
Informal private betting	4.07 *** (2.66; 6.23)	1.91 *** (1.42; 2.56)	4.15 *** (2.68; 6.42)	3.67 *** (2.73; 4.95)	5.82 *** (3.73; 9.09)	3.04 *** (2.27; 4.07)	4.36*** (2.79; 6.82)	2.76 *** (1.99; 3.84)
Total monetary gambling activities ^a^	0.40 ***	0.19 ***	0.42 ***	0.43 ***	0.43 ***	0.45 ***	0.43 ***	0.34 ***

Note: ^a^ Associations between total monetary gambling activities and each simulated gambling activity are reported as Pearson’s *r* coefficients. * *p* < 0.05; ** *p* < 0.01; *** *p* < 0.001.

**Table 3 ijerph-19-10652-t003:** Past-month participation in each form of simulated gambling predicting problematic gambling controlling for adolescent age, gender, impulsivity and involvement in monetary gambling in past month.

	Model 1 Controls			Model 2 Monetary Gambling			Model 3 Games with ‘Mini’ Gambling Components			Model 4 Casino Games via Apps			Model 5 Casino Games on Social Networking			Model 6 Free Demo Games		
	*B*	*SE*	β	*B*	*SE*	β	*B*	*SE*	β	*B*	*SE*	β	*B*	*SE*	β	*B*	*SE*	β
Qualtrics sample (*N* = 826)																		
Age	0.23	0.04	0.19 ***	0.11	0.03	0.08 **	0.11	0.03	0.09 ***	0.09	0.03	0.08 **	0.10	0.03	0.08 **	0.10	0.03	0.08 **
Gender	−0.43	0.14	−0.11 **	−0.18	0.11	−0.04	−0.13	0.11	−0.03	−0.17	0.11	−0.04	−0.14	0.11	−0.04	−0.14	0.11	−0.04
Impulsivity	0.04	0.02	0.09 **	0.05	0.01	0.11 ***	0.05	0.01	0.11 ***	0.05	0.01	0.11 ***	0.05	0.01	0.10 ***	0.05	0.01	0.10 ***
Monetary gambling (sum of activities)				0.59	0.03	0.63 ***	0.53	0.03	0.57 ***	0.54	0.03	0.58 ***	0.53	0.03	0.57 ***	0.50	0.03	0.53 **
Simulated gambling activity							0.90	0.15	0.17 ***	−0.75	0.16	0.13 ***	0.92	0.17	0.16 ***	1.37	0.16	0.23 ***
	*R*^2^ = 0.05, *F*(3, 822) = 15.31, *p* < 0.001			*R*^2^ = 0.44, *F*(4, 821) = 160.11, *p* < 0.001			*R*^2^ = 0.46, *F*(5, 820) = 140.41, *p* < 0.001			*R*^2^ = 0.46, *F*(5, 820) = 135.66, *p* < 0.001			*R*^2^ = 0.46, *F*(5, 820) = 138.51, *p* < 0.001			*R*^2^ = 0.48, *F*(5, 820) = 152.87, *p* < 0.001		
Email/Ads sample (*N* = 843)																		
Age	0.09	0.05	0.06	−0.01	0.03	−0.00	−0.08	0.03	−0.00	−0.01	0.03	−0.00	−0.00	0.03	−0.00	−0.00	0.03	−0.00
Gender	0.33	0.19	0.06	0.07	0.12	0.01	0.08	0.12	0.01	0.04	0.12	0.01	0.04	0.12	0.01	0.06	0.12	0.01
Impulsivity	0.21	0.02	0.30 ***	0.09	0.01	0.14 ***	0.09	0.01	0.14 ***	0.09	0.01	0.13 ***	0.08	0.01	0.12 ***	0.09	0.01	0.13 ***
Monetary gambling (sum of activities)				1.02	0.03	0.76 ***	1.01	0.03	0.76 ***	0.97	0.03	0.73 ***	0.95	0.03	0.72 ***	0.99	0.03	0.75 ***
Simulated gambling activity							0.09	0.12	0.02	0.53	0.13	0.09 ***	0.63	0.13	0.11 ***	0.32	0.14	0.05 *
	*R*^2^ = 0.10, *F*(3, 839) = 29.98, *p* < 0.001			*R*^2^ = 0.65, *F*(4, 838) = 385.15, *p* < 0.001			*R*^2^ = 0.65, *F*(5, 837) = 308.04, *p* < 0.001			*R*^2^ = 0.66, *F*(5, 837) = 317.56, *p* < 0.001			*R*^2^ = 0.66, *F*(5, 837) = 321.32, *p* < 0.001			*R*^2^ = 0.65, *F*(5, 837) = 310.62, *p* < 0.001		

* *p* < 0.05; ***p* < 0.01; ****p* < 0.001.

**Table 4 ijerph-19-10652-t004:** Monetary expenditure and time spent playing on simulated gambling games as risk factors for gambling problems among simulated gamblers.

	Model 1 Controls			Model 2 Expenditure in Simulated Gambling			Model 3 Time Spent in Simulated Gambling			Model 4 Expenditure and Time Spent		
Qualtrics Sample (*N* = 339)												
	*B*	*SE*	β	*B*	*SE*	β	*B*	*SE*	β	*B*	*SE*	β
Age	0.27	0.09	0.17 **	0.08	0.08	0.05	0.19	0.08	0.12 *	0.08	0.07	0.05
Gender	−0.20	0.27	−0.04	0.15	0.24	0.03	0.18	0.25	0.04	0.30	0.23	0.06
Impulsivity	0.04	0.03	0.07	0.00	0.03	0.00	0.07	0.03	0.12 *	0.03	0.03	0.05
Expenditure in simulated gambling (log)				1.73	0.16	0.52 ***	--	--	--	1.33	0.17	0.40 ***
Time spent (hours) in simulated gambling in past month (log)							2.10	0.23	0.45 ***	1.29	0.24	0.28 ***
	*R*^2^ = 0.03, *F*(3, 335) = 3.84, *p* = 0.01			*R*^2^ = 0.29, *F*(4, 334) = 33.43, *p* < 0.001			*R*^2^ = 0.22, *F*(4, 334) = 23.96, *p* < 0.001			*R*^2^ = 0.35, *F*(5, 333) = 35.04, *p* < 0.001		
Email/Ads sample (*N* = 636)												
Age	0.04	0.06	0.03	0.01	0.04	0.01	−0.03	0.05	−0.02	−0.01	0.04	−0.01
Gender	0.50	0.21	0.09 *	0.11	0.16	0.02	0.55	0.19	0.10 **	0.17	0.15	0.03
Impulsivity	0.13	0.03	0.18 **	0.03	0.02	0.04	0.09	0.03	0.13 ***	0.03	0.02	0.04
Expenditure in simulated gambling (log)				1.93	0.08	0.68 ***	--	--	--	1.72	0.09	0.61 ***
Time spent (hours) in simulated gambling in past month (log)							3.06	0.26	0.42 ***	1.26	0.23	0.17 ***
	*R*^2^ = 0.04, *F*(3, 632) = 8.88, *p* < 0.001			*R*^2^ = 0.48, *F*(4, 631) = 144.13, *p* < 0.001			*R*^2^ = 0.21, *F*(4, 631) = 42.70, *p* < 0.001			*R*^2^ = 0.50, *F*(5, 630) = 126.84, *p* < 0.001		

* *p* < 0.05; ** *p* < 0.01; *** *p* < 0.001.

## Data Availability

Data from this study can be requested from the NSW Office of Responsible Gambling.

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
