# Peer review of "Adolescents Who Play and Spend Money in Simulated Gambling Games Are at Heightened Risk of Gambling Problems"

_ijerph, 2022, doi:10.3390/ijerph191710652_

Round 1

Reviewer 1 Report

In this paper, the authors present a detailed analysis for Adolescents who Play and Spend Money in Simulated Gambling Games. The results are corroborative and interesting. I can recommend it for a publication after several concerns are properly addressed.  

1.    The topic should be revised to more reflect research problem.

2.   The authors should highlight the advantage or novelty of their work by discussing main differences with previous literatures

3.     The main contributions should be summarized in the introduction.

4.  The authors use too many tables. If possible, I also recommend the authors to replace part of them with graphic representation.

Author Response

In this paper, the authors present a detailed analysis for Adolescents who Play and Spend Money in Simulated Gambling Games. The results are corroborative and interesting. I can recommend it for a publication after several concerns are properly addressed.

Response: Thank you for your positive comments.

  1. The topic should be revised to more reflect research problem.

Response: We have now clarified the research problem in the first paragraph.

  1. The authors should highlight the advantage or novelty of their work by discussing main differences with previous literatures

Response: When discussing previous studies, we have have now added: “However, previous studies examining relationships between simulated gambling and problem gambling have relied on small sub-samples of at-risk/problem gamblers, have not always used a validated measure of problem gambling, have lowered the cut-off score of validated instruments to have sufficient numbers for analysis, or have included only one type of simulated gambling. The current study was designed to overcome these limitations.”

  1. The main contributions should be summarized in the introduction.

Response: We consider that we should not pre-empt the study’s findings by identifying its exact contributions in the introduction. However, after explaining the hypotheses, we have added “The study will contribute to knowledge by testing these hypotheses, which in turn can inform public health measures to protect young people from any risks to their wellbeing associated with playing simulated gambling games.”

  1. The authors use too many tables. If possible, I also recommend the authors to replace part of them with graphic representation.

 Response: There are 4 tables, which is fewer than many quantitative papers. Unfortunately, we cannot see how we can reduce the number of tables without losing essential information that provides evidence of the findings, nor how this material could be presented in a graphic representations.

Reviewer 2 Report

The manuscript has a relevant topic for public health. It is well written and easy to read. The aim of the study was to explore the associations between youth participation in simulated gambling and the risk of gambling problems. The findings of this study are important globally.

There are some minimal comments that I believe might increase the quality of the manuscript:

In the abstract, the mean age of two samples might be included.

According to the requirements of the journal, the citation should be numerical.

Subsection 1.1. presents not only types of gambling, but also the prevalence of gambling in each of them. The name of the subsection might be revised accordingly.

In what units “expenditure in simulated gambling in the past month” (Table 1) was measured?

What possible answers were for assessing the Monetary expenditure on simulated gambling games (lines 259-262)?

Line 194. Is it correct to state “the most participants” if the percentage is 55.1?

For demographic characteristics’ description, please insert (see Table 1), line 267.

For Statistical Analysis please insert information on what statistical tool (SPSS?) was used for statistics.

In discussion, I would suggest starting straightly from the main question of the study and describing what new this study adds. I have doubts that information in lines 405-411 is needed.

Conclusions might be more concrete. The study has important practical implications. The information in lines 528-546 might be a subsection in the discussion section “Practical implications”. In conclusions, this information might be presented but in a shortened way.

Author Response

The manuscript has a relevant topic for public health. It is well written and easy to read. The aim of the study was to explore the associations between youth participation in simulated gambling and the risk of gambling problems. The findings of this study are important globally.

Response: Thank you for your positive comments.

There are some minimal comments that I believe might increase the quality of the manuscript:

In the abstract, the mean age of two samples might be included.

Response: We have now included the mean age of both samples in the abstract.

According to the requirements of the journal, the citation should be numerical.

Response: We have now used the numerical system.

Subsection 1.1. presents not only types of gambling, but also the prevalence of gambling in each of them. The name of the subsection might be revised accordingly.

Response: We have now renamed this sub-section “Types and prevalence of simulated gambling”

In what units “expenditure in simulated gambling in the past month” (Table 1) was measured?

What possible answers were for assessing the Monetary expenditure on simulated gambling games (lines 259-262)?

Response: We have now clarified in the measures section and Table that expenditure was reported in Australian dollars (AUD).

Line 194. Is it correct to state “the most participants” if the percentage is 55.1?

Response: We have now changed this to “just over half of participants”.

For demographic characteristics’ description, please insert (see Table 1), line 267.

Response: We have now inserted “see Table 1” where demographics are described.

For Statistical Analysis please insert information on what statistical tool (SPSS?) was used for statistics.

Response: This information has now been included in the first sentence of the Statistical Analyses section.

In discussion, I would suggest starting straightly from the main question of the study and describing what new this study adds. I have doubts that information in lines 405-411 is needed.

Response: We have now removed lines 405-411. We have now started the discussion so it immediately refers to the discussion of results of the hypotheses testing.

Conclusions might be more concrete. The study has important practical implications. The information in lines 528-546 might be a subsection in the discussion section “Practical implications”. In conclusions, this information might be presented but in a shortened way.

Response: We have now moved some of the previous conclusions into a separate sub-section in the discussion, titled Practical implications. We have also made these implications more concrete. As a result, the conclusion is now shorter.